# Evaluation of the Relative Phase Comparison Method at Its Limits Used for Absolute Phase Determination of an AC Cage Current Shunt Group

**DOI:** 10.3390/s23208510

**Published:** 2023-10-17

**Authors:** Stipe Gašparini, Petar Mostarac, Jure Konjevod, Roman Malarić

**Affiliations:** 1KONČAR-Electrical Engineering Institute Ltd., 10002 Zagreb, Croatia; 2Faculty of Electrical Engineering and Computing, University of Zagreb, 10000 Zagreb, Croatia; petar.mostarac@fer.hr (P.M.); jure.konjevod@fer.hr (J.K.); roman.malaric@fer.hr (R.M.)

**Keywords:** AC current shunts, cage design, phase characteristics, relative comparison method, absolute phase verification, phase error

## Abstract

AC current shunts are used for precise current measurements. The application of AC current shunts requires that their amplitude phase characteristics are known. A group of three geometrically identical current shunts and a reference shunt are observed in this paper. The phase characteristics of the reference shunt have been previously obtained. A relative phase comparison has been made between the three geometrically identical shunts, and phase displacement values for each have been obtained. After this, the results are verified with the reference shunt. The relative method is most suitable for shunts, where their respective *RC* and *L/R* values are small (compared with 1/ω) and of the same order. The ratios of the nominal resistance values of the shunts used in this paper are at the limit of the given statement. The conclusion is that the method applied at the mentioned limits, in terms of the metrology-grade phase angle determination of current shunts, is not to be considered reliable at frequencies higher than 1 kHz.

## 1. Introduction

The continuous development and use of measuring equipment intended to measure AC currents and related electric power requires precise measurement methods as well as referent standards with known characteristics. Accordingly, metrology-grade current transducers, i.e., current shunts, are usually applied in such measurements [1,2,3]. For a DC current shunt, the most important value is its resistance. On the other hand, the main quantities of interest for an AC current shunt are its inductance and capacitance. Thus, an AC current shunt is defined by its impedance, which can be expressed as $Z=\left|Z\right|e^{i\varphi }$, where $\varphi =\arg\left(Z\right)$. For the comprehensive application of AC current shunts, it is required that their amplitude phase characteristics are previously determined. The cage current shunts [4,5] investigated in this paper are constructed of double-sided FR4 fiberglass-epoxy PCB material, three circular PCB elements, crossbars and resistors. The current flows from the input side of the shunt through the crossbars, each carrying a fraction of the input current to the resistors connected in parallel at the output side. The voltage drop that occurs on the resistors is measured at the output side of the shunt. A commonly used type of these resistors is manufactured by Vishay, characterized by the use of metal film technology. They have a very low temperature coefficient, as well as inductance and capacitance values. The design requirements of AC current shunts can be summarized as follows [6,7]:Small capacitance for low current ranges;Small inductance for high current ranges;Coaxial structure;Good heat-conduction path from the resistive elements to the surface;Sufficient surface area for heat dissipation;An influence of the skin effect that is as small as possible;Small drift, i.e., good stability.

Recent work on the development of AC current shunts has focused on increasing their nominal current values and their bandwidth capabilities, such as in [8,9,10,11].

This paper focuses on the determination of the phase characteristics of cage current shunts. Several methods can be used to determine the phase characteristics of current shunts [12,13,14,15]. Some of the recent research papers cite methods such as using vector network analyzers to obtain the phase characteristics of current shunts [16] as well as using the Josephson voltage standard [17]. The focus of this paper is to evaluate the relative phase comparison method proposed in [12] at its limits, regarding:The ratios of the nominal current;The resistance values of shunts used in the comparison.

The method shown in [12] has been chosen because it offers a simplified approach to the determination of phase characteristics of AC current shunts compared to other methods, and as such is interesting for use in evaluating its limits and accuracy. The relatively simple approach of the method also makes it interesting for commercial measurements, where the highest level of accuracy is not necessary. A group of three geometrically identical current shunts with nominal current values of 0.1 A, 0.3 A and 1 A is observed in this paper. In addition to that, a reference current shunt with the nominal current value of 1 A is used. The three geometrically identical cage current shunts were previously developed at FER Zagreb [18]. An example of the current shunt can be seen in Figure 1. The reference current shunt is a model A40B-1A cage current shunt manufactured by Fluke for which the absolute phase displacement values are known. The A40B-1A has been calibrated for absolute phase displacement values in the frequency range from 50 Hz to 10 kHz.

## 2. Procedure for Evaluation of the Relative Phase Comparison Method

In the first phase of the evaluation, a relative phase comparison was undertaken using the method proposed in [12] using the three shunts developed at FER, and absolute phase displacement values for each shunt have been obtained. In [12], a statement is made that the proposed method is most suitable for shunts, where their respective RC and L/R are small (compared with 1/ω) and of the same order. The ratios of the nominal resistance and current values of the three geometrically identical shunts used in this paper are larger than those of the groups observed in [12,15], and are at the limit of the given statement. The main goal of this paper is to evaluate the relative phase comparison method at the given limit.

Nominal resistance values of the shunts used in this paper, as well as the ones from papers [12,15], are shown in Table 1. The third row in the table shows the resistance and nominal current values of the shunts used in this paper. The highest ratio of the nominal resistance and current values between the shunts from [12] is 2.5, the ratio from [15] is 5, while the ratio in this paper is 10, which is in fact the limit of the statement mentioned above.

In the second phase of the evaluation, a phase comparison is made between the 1 A shunt and the reference current shunt. Consequently, mutual comparisons of the measurement results obtained in each way are presented further in the paper, and relevant data are accordingly analyzed and discussed. This set of measurements marks the first time the absolute phase of the cage current shunts developed at FER has been determined. Figure 2 shows a block diagram of the explained analysis.

## 3. Measurement Setup and Method

### 3.1. Measurement Setup

The overall measurement setup for the realization of used measurement methods is based on an NI PXI 1062Q chassis with the related data acquisition module NI PXI 4461. Regarding power supply, a combination of a function generator RIGOL DG4062 and a power amplifier Toellner TOE-7621 has been utilized [19,20,21]. Accordingly, signal generation is achieved with a function generator DG4062, which provides the voltage input signal necessary for the amplifier’s input (CTRL IN—12 V max). In the next step, the amplified signal is forwarded from the amplifier’s output (OUT) to two current shunts under test connected in series with a Tee connector [22], as depicted on Figure 3. Output voltages from the current shunts are forwarded to the analog input channels AI 1 and AI 2 of the NI 4461 PXI module for sampling. The NI PXI 4461 digitizer has two simultaneously sampled input channels with a 24-bit analog-to-digital converter (ADC) and can perform data acquisition with a sampling rate of up to 204.8 kSa/s [23].

### 3.2. Measurement Method for Phase Difference Determination

#### 3.2.1. Introduction to the Measurement Method

To determine the absolute phase, the method proposed in [12] was used. The software program developed in LabVIEW 2019 serves for the automatization of the whole measurement and data acquirement procedure. The mutual relative phase comparison between three current shunts with nominal currents of 0.1 A, 0.3 A and 1 A has been done to determine the absolute phase of each shunt. The substantial prerequisite of the proposed method assumes that all three shunts are of equal dimensions, i.e., their inductance L and capacitance C are identical [12]. Another condition that needs to be satisfied is that the effective current must be constant across the three shunts, i.e., the current was set to 0.1 A in each mutual comparison between the two shunts.

#### 3.2.2. Sampling Rate and Frequency Range

The frequency of the input signal was set across a specified range, i.e., from 50 Hz to 10 kHz. The DFT algorithm was applied to the acquired data to calculate relative phase differences between two current shunts in each step of the measurement method. Since the DFT algorithm, without any window function, is not immune to incoherent sampling, as elaborated in [24], the sampling rates and number of samples taken were varied, respectively, to ensure that the exact number of integer periods of the input signal was analyzed. The measurement results were obtained using the sampling frequencies shown in Table 2, chosen to ensure that the phase between each pair of shunts would be calculated using the same number of periods. For each frequency, 10 periods were used to obtain the phase difference.

#### 3.2.3. Repeatability and Influence of Data Length (Number of Periods)

Measurements were also made to evaluate whether the difference in the number of periods used in the measurement influences the measurement results. At the 1 kHz frequency, measurements were repeated using the sampling rate of 200 kSa/s with 20,000 samples taken (200 periods), but there was no significant difference in the results. Additionally, for the 10 kHz frequency, measurements were repeated for the sampling rate of 200 kSa/s with 200 samples (10 periods), 2000 samples (100 periods), 20,000 samples (1000 periods) and 200,000 samples (10,000 periods). A significant difference in the results was not noticed. Besides this, measurements were repeated four months apart, and the results also confirm the repeatability of the measurements.

As already mentioned, the NI PXI-4461 module was used as a two-channel digitizer [25]. Output voltages from the shunts were simultaneously sampled using an internal 10 MHz clock. Thus, the acquired voltages were processed with the developed LabVIEW application.

#### 3.2.4. Phase Analysis from Relative Phase Comparison Method

The impedance phase angle of the current shunt is defined with the following expression:(1)$$\varphi =\arctan\left(\frac{Im\left[Z_{\mathrm{sh}}\right]}{Re\left[Z_{\mathrm{sh}}\right]}\right),$$
where $Im\left[Z_{\mathrm{sh}}\right]$ and $Re\left[Z_{\mathrm{sh}}\right]$ are the imaginary and real parts of the shunt impedance $Z_{\mathrm{sh}}$. The current shunt and its related input and output connectors can be modeled in various ways, such as in [26]. However, a simplified and more precise model that can be applied for phase difference determination has been elaborated in [2]. A simplified model from [27] is presented in Figure 4.

The simplified shunt model from [27] is defined as an ordinary load with high input resistance. The mutual inductance $M_{\mathrm{SH}}$ along with the shunt inductance $L_{\mathrm{SH}}$ defines the total inductance L. The total capacitance C can be defined by the shunt’s capacitance $C_{\mathrm{SH}}$, the capacitance introduced to the system with the connecting cables $C_{\mathrm{CB}}$ and the capacitance of the voltage measurement device that follows the shunt $C_{\mathrm{IN}}$. The transfer function of the model is then equal to:(2)$$Z=\frac{R-\omega^{2}RLC-j\omega R^{2}C+j\omega L-j\omega^{3}L^{2}C+\omega^{2}RLC}{{\left(1-\omega^{2}LC\right)}^{2}+{\left(\omega RC\right)}^{2}}.$$

Since all ωC and ωL terms are smaller than R for the frequencies of interest, they can be disregarded, and so too can squared components [27]. Further, the squared ωRC component is also not significant, and is disregarded. Finally, the related transfer function can be expressed as
(3)$$Z\approx R+j\omega L_{\mathrm{eq}},$$
where $L_{\mathrm{eq}}$ is the equivalent inductance, defined as
(4)$$L_{\mathrm{eq}}=L-R^{2}C.$$

From Equations (1) and (3), the following expressions can be derived:(5)$$\varphi_{i}=\arctan\left(\frac{\omega L_{\mathrm{eq}}}{R_{i}}\right)\approx \frac{\omega L_{\mathrm{eq}}}{R_{i}},$$
(6)$$\varphi_{i}=\frac{\omega \left(L-R_{i}^{2}C\right)}{R_{i}}.$$

Since current shunts are expected to have small phase angle errors, the phase $\varphi_{i}$ of a shunt $R_{i}$ can be effectively approximated with:(7)$$\varphi_{i}=\frac{\omega L}{R_{i}}-\omega R_{i}C.$$

Consequently, phase differences between each pair of current shunts can be written as:(8)$$\phi_{12}=\varphi_{1}-\varphi_{2},$$
(9)$$\phi_{23}=\varphi_{2}-\varphi_{3},$$
(10)$$\phi_{31}=\varphi_{3}-\varphi_{1}.$$

Combining (7) with (8), (9) and (10), the phase differences between each pair of current shunts can be written as:(11)$$\phi_{12}=\omega \left[L\left(\frac{1}{R_{1}}-\frac{1}{R_{2}}\right)-C\left(R_{1}-R_{2}\right)\right],$$
(12)$$\phi_{23}=\omega \left[L\left(\frac{1}{R_{2}}-\frac{1}{R_{3}}\right)-C\left(R_{2}-R_{3}\right)\right],$$
(13)$$\phi_{31}=\omega \left[L\left(\frac{1}{R_{3}}-\frac{1}{R_{1}}\right)-C\left(R_{3}-R_{1}\right)\right].$$

Solving these equations for ωL and ωC using the measured phase differences $\phi_{12}$, $\phi_{23}$ and $\phi_{31}$, as well as the known DC resistance values $R_{1}$, $R_{2}$ and $R_{3}$, three values of ωL and ωC can be obtained. After this, an average value for ωL and ωC is taken and the absolute phase displacement value for each shunt can be obtained using Equation (7).

Combining Equations (8)–(10), the following expression can be derived:(14)$$\phi_{12}+\phi_{23}+\phi_{31}=0.$$

Ideally, the sum of the three measured phase differences is equal to 0. However, in real measurements, the mentioned sum is in the order of 1 µrad for, e.g., a frequency of 50 Hz, and increases up to the order of 100 µrad at frequencies higher than 1 kHz [12].

### 3.3. Comparison against Reference Shunt

To evaluate the results obtained using the relative phase comparison method, a phase comparison was made using a reference current shunt for which the absolute phase displacement had been previously obtained. The measurement setup for this method is the same as the one used in the previous paragraph. The only difference between the two setups is that during this measurement, a shunt of unknown phase displacement was compared with the reference shunt of known phase displacement. The obtained phase difference value can be simply defined as:(15)$$\phi_{iR}=\varphi_{i}-\varphi_{R},$$
where phase $\varphi_{i}$ is the unknown phase displacement value and $\varphi_{R}$ is the reference shunt phase displacement.

### 3.4. Phase Measurement Error of the Measurement System

The phase measurement error of the measurement system is defined as the phase measurement error between the two channels of the NI PXI-4461. The measured phase difference between a pair of two shunts can be represented by the following equation:(16)$$\phi_{12}=\varphi_{1}-\varphi_{2}+\phi ,$$
where $\phi_{12}$ is the measured phase difference, $\varphi_{1}$ and $\varphi_{2}$ are the absolute phase displacement values of the two shunts and ϕ is the phase difference measurement error between the two channels of the NI PXI-4461.

To obtain the phase measurement error ϕ in this specific case, three sets of measurements were made. The first one was between the 0.3 A and 1 A FER shunts (used in the relative phase comparison). The second one was between the reference Fluke 1 A shunt and the 1 A FER shunt. Finally, the third one was made between the 0.3 A FER shunt and the reference Fluke 1 A shunt. Using Equation (16), the following three equations can be defined:(17)$$\phi_{23}=\varphi_{2}-\varphi_{3}+\phi ,$$
(18)$$\phi_{ref3}=\varphi_{ref}-\varphi_{3}+\phi ,$$
(19)$$\phi_{2ref}=\varphi_{2}-\varphi_{ref}+\phi ,$$
where $\phi_{23}$ is the measured phase difference between the 0.3 A and 1 A shunts, $\phi_{ref3}$ is the measured phase difference between the reference and the 1 A shunt, while $\phi_{2ref}$ is the measured phase difference between the 0.3 A and reference shunts. Combining the three equations, the value of ϕ can be calculated using the following equation:(20)$$\phi =\phi_{2ref}-\phi_{23}+\phi_{ref3}.$$

The calculated phase error values obtained during the measurements for frequencies of interest are shown in Table 3.

The calculated phase error values were considered during the phase difference measurements and applied to the measurement results as corrections. The measurement results obtained using the relative phase comparison method and the absolute phase verification method with the reference shunt are presented in the following chapter.

## 4. Measurement Results and Discussion

### 4.1. Relative Phase Comparison

The phase displacement values of the three current shunts manufactured at FER, obtained using the relative phase comparison method, are shown in Figure 5, Figure 6 and Figure 7. The obtained phase displacement values range from 0 µrad at 50 Hz up to −866 µrad at 10 kHz for the 0.1 A shunt, from 1 µrad at 50 Hz up to −255 µrad at 10 kHz for the 0.3 A shunt, and from 2 µrad at 50 Hz up to −70 µrad at 10 kHz for the 1 A shunt.

### 4.2. Absolute Phase Verification with Reference Current Shunt

Phase comparison using the absolute phase verification method was performed between the 1 A shunt used in the relative comparison method and the reference 1 A shunt. The main condition when using the absolute phase verification method is the predetermined phase characteristics of the reference current shunt. Phase displacement values with the corresponding measurement uncertainty for the calibrated reference shunt are given in Table 4.

### 4.3. Results Comparison

Since the absolute phase displacement values of the 1 A FER shunt were obtained using both methods, a comparison of the results is possible. Table 5 shows the phase displacement values obtained for the 1 A shunt using both the relative comparison method and the absolute phase verification method.

The absolute phase error between the two methods for the 1 A shunt is given in Table 6.

The same comparison can also be made for the other two current shunts. Figure 8, Figure 9 and Figure 10 shown below illustrate a comparison of the results for all three shunts derived via both methods.

It can be seen from the figures shown above that the results obtained via the absolute and relative comparison methods practically overlap up to the frequency of 1 kHz. Consequently, the mutual difference between results obtained with the two methods is very small up to 1 kHz, and in this frequency range, the relative phase comparison method can be considered reliable even at its limits in terms of the highest ratio of the nominal resistance and current values between the shunts, as elaborated in Section 2 and presented in Table 1. However, at higher frequencies, this difference increases proportionally with frequency. Thus, the method investigated at the mentioned limits and in terms of the metrology-grade phase angle determination of current shunts is not to be considered as reliable at frequencies higher than 1 kHz. The measurement uncertainty budgets of the relative comparison method and the absolute phase verification method are analyzed in the following chapter.

## 5. Uncertainty Analysis

### 5.1. Relative Phase Comparison Method

The main factors that determine the measurement uncertainty of the relative phase comparison measurement method are the stability of the measurement system, i.e., the stability of the PXI-4461 (type A evaluation), and the PXI-4461 phase measurement uncertainty (type B evaluation) of the calculated error between its two channels.

The stability $u_{A}$ of the system and its measurement uncertainty contribution is defined by the average standard deviation of the measurements. The relative phase comparison between the three shunts was repeated twice, four months apart, to ensure repeatability. Each measurement consisted of 10 measurement results for each frequency of interest. The worst-case average standard deviation obtained via the phase difference measurements for each pair of shunts has been used to determine the measurement uncertainty. The average standard deviation was acquired using the equations
(21)$$s\left(\bar{x}\right)=\frac{s\left(q_{k}\right)}{\sqrt{n}},$$
and
(22)$$s^{2}\left(q_{k}\right)=\frac{1}{n-1}\sum_{k=1}^{n}{\left(q_{k}-\bar{q}\right)}^{2},$$
where n is the number of measurements (n = 20), $q_{k}$ represents each phase difference measurement value and $\bar{q}$ is their average.

The phase measurement uncertainty of the PXI-4461 $u_{B}$ is defined with the measurement uncertainty of the calculated phase measurement error between its two channels $u\left(\phi \right)$, which is defined in Section 3.4 using Equation (20).

The measurement uncertainty of the calculated phase measurement error $u\left(\phi \right)$ is determined by the measurement uncertainty of each of the three phase difference values used to calculate it: $u\left(\phi_{2ref}\right)$*,* $u(\phi_{23})$ and $u(\phi_{ref3})$. The measurement uncertainty contribution $u\left(\phi_{2ref}\right)$ is defined by the average standard deviation of the $\phi_{2ref}$ measurements and the uncertainty associated with the reference 1 A shunt taken from its calibration certificate (Table 4), given for *k* = 2. The same logic applies to the $u(\phi_{ref3})$ contribution. The measurement uncertainty contribution $u\left(\phi_{23}\right)$ is defined by the average standard deviation of the $\phi_{23}$ measurements. Finally, the measurement uncertainty of the phase measurement error $u\left(\phi \right)$ for *k* = 1 is determined using the following equation:(23)$$u\left(\phi \right)=\sqrt{u^{2}\left(\phi_{2ref}\right)+u^{2}\left(\phi_{23}\right)+u^{2}\left(\phi_{ref3}\right)}.$$

The calculated measurement uncertainty $u\left(\phi \right)$ is given in Table 7.

Finally, the total expanded measurement uncertainty values for the relative phase comparison method calculated using its two components $u_{A}$ and $u_{B}$ are given in Table 8.

### 5.2. Absolute Phase Verification Method

The measurement uncertainty of the absolute phase verification method is defined by three main components. The first component refers to the phase measurement uncertainty associated with the PXI-4461, which is shown in the previous section for the relative method. The second one is the measurement uncertainty associated with the reference shunt phase displacement, which has been taken from the calibration certificate. Finally, the third component is defined by the average standard deviation of the phase difference measurements. Taking all of this into account using the approach given in Section 5.1, the expanded measurement uncertainty is given in Table 9.

## 6. Conclusions and Discussion

The main goal of this paper is to evaluate the relative phase comparison method at its limits regarding the nominal current and resistance values of the current shunts used. The results obtained using the relative phase comparison method have been compared with the results obtained using the absolute phase verification method, using a reference current shunt of known phase displacement. The analysis of the obtained results concludes that the results obtained by the absolute and relative phase comparison methods practically overlap up to the frequency of 1 kHz. Consequently, the difference between the results obtained with the two methods is very small up to 1 kHz, and in this frequency range, the relative comparison method can be considered reliable even at its limits in terms of the highest ratio of the nominal resistance and current values between the shunts. However, at higher frequencies, the difference between the two methods increases. Since the measurement error of the PXI, in other words the error of the measurement setup, has been calculated and considered as a correction factor for the measurements, it is not the main factor contributing to the error between the two methods. As stated previously in the paper wherein the method was introduced, ideally, the sum of the phase differences of the shunts is equal to 0. In real measurements, the mentioned sum is in the order of 1 µrad for, e.g., a frequency of 50 Hz, and increases up to an order of 100 µrad at frequencies higher than 1 kHz. In the case of this paper, this sum at 10 kHz is equal to −659 µrad. Since this is the case, it can be concluded that the accuracy of the method drops when the ratios of the nominal values of the shunts are of the order investigated in this paper. The measurement setup has shown an influence on the measurement results, but it is not the main factor causing the error between the two methods. Future work will include evaluating the method with a higher-accuracy PXI system. To summarize, applying the investigated method at the mentioned limits and in terms of the metrology-grade phase angle determination of current shunts is not to be considered as reliable at frequencies higher than 1 kHz.

## Figures and Tables

**Figure 1 sensors-23-08510-f001:**
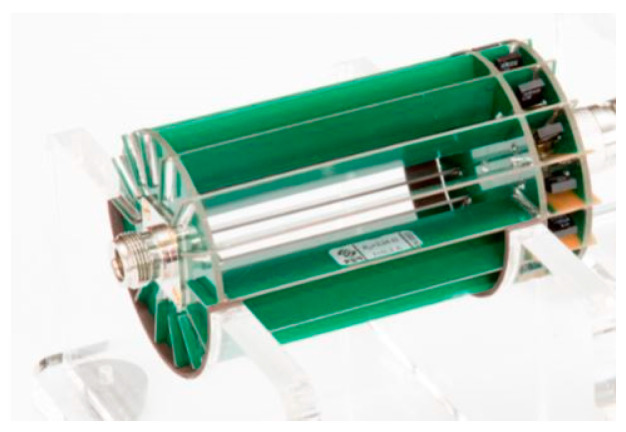
FER cage current shunt.

**Figure 2 sensors-23-08510-f002:**
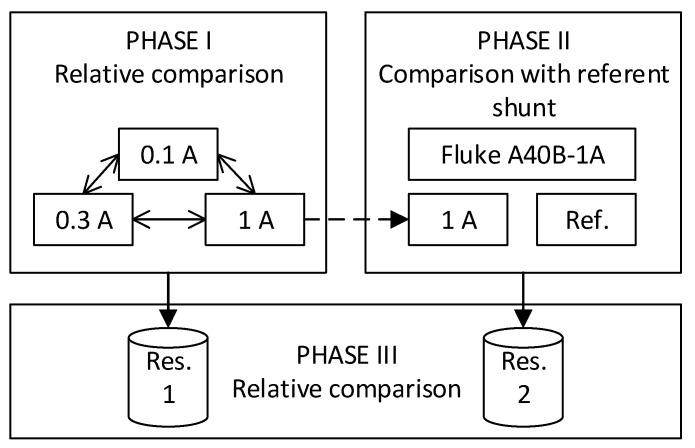
Analysis block diagram.

**Figure 3 sensors-23-08510-f003:**
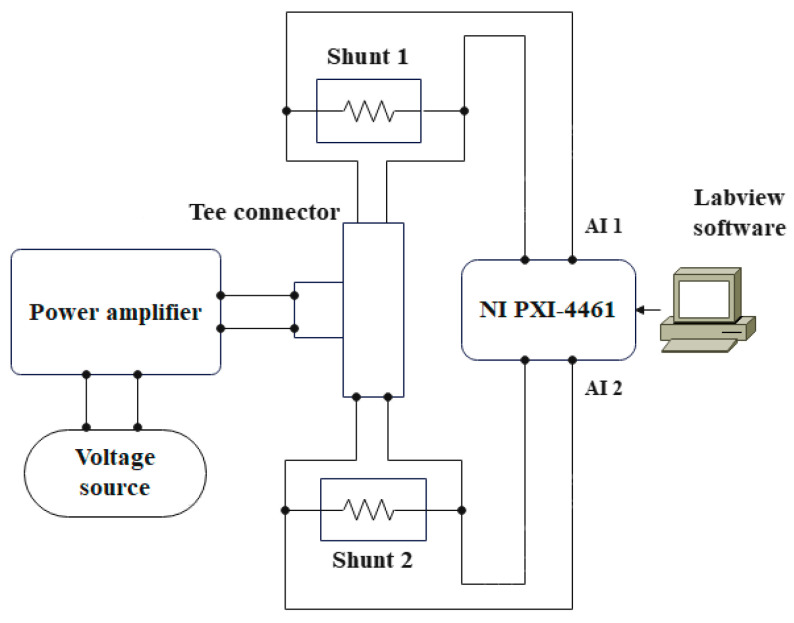
Measurement setup.

**Figure 4 sensors-23-08510-f004:**
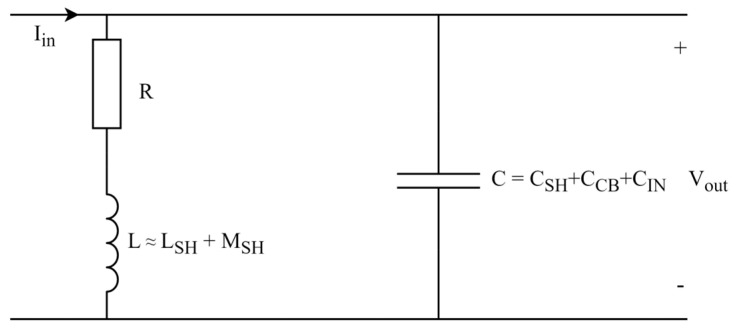
Simplified model of the current shunt [27].

**Figure 5 sensors-23-08510-f005:**
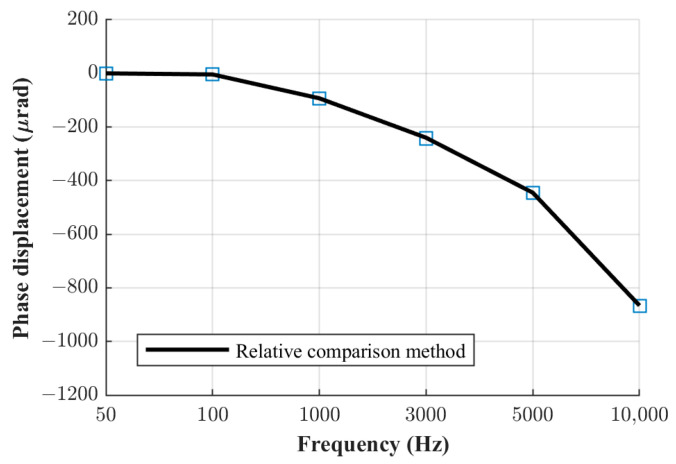
Phase displacement for 0.1 A shunt.

**Figure 6 sensors-23-08510-f006:**
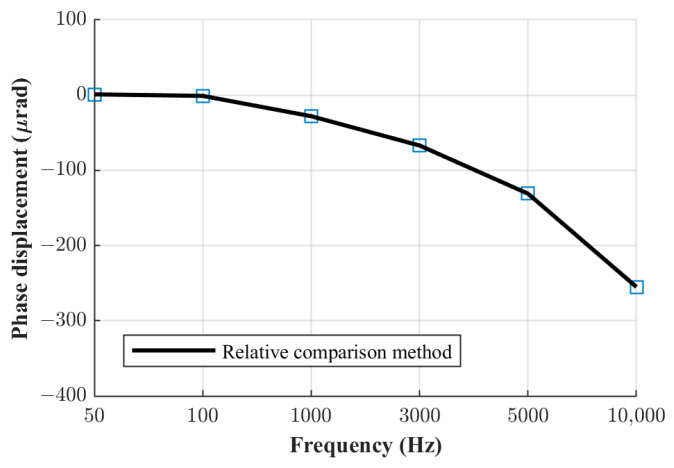
Phase displacement for 0.3 A shunt.

**Figure 7 sensors-23-08510-f007:**
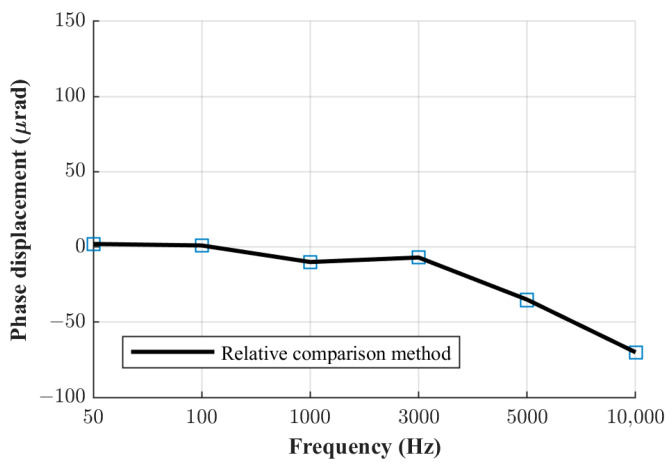
Phase displacement for 1 A shunt.

**Figure 8 sensors-23-08510-f008:**
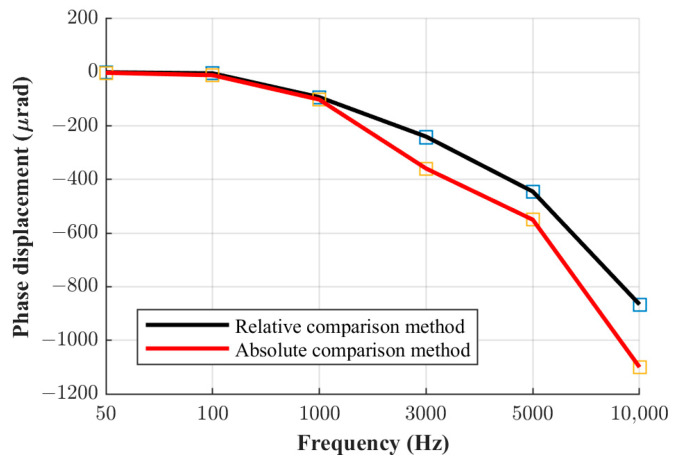
Phase displacement for the 0.1 A shunt.

**Figure 9 sensors-23-08510-f009:**
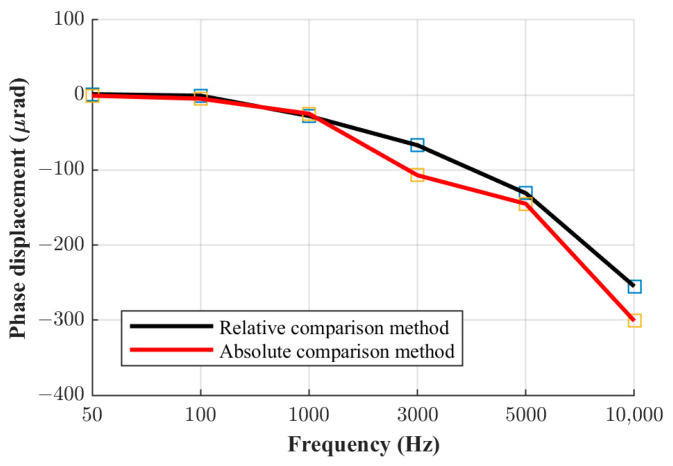
Phase displacement for the 0.3 A shunt.

**Figure 10 sensors-23-08510-f010:**
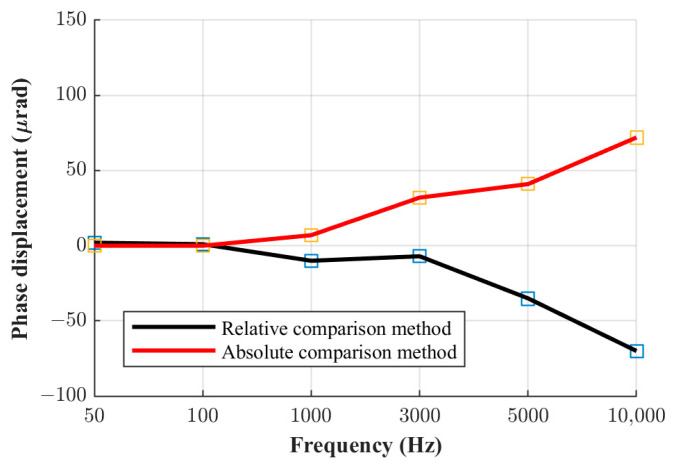
Phase displacement for the 1 A shunt.

**Table 1 sensors-23-08510-t001:** Nominal shunt resistance values.

Shunt Group	Nominal Resistance/Current Values		
RISE [12]	$R_{1}=0.4\;$Ω $I_{1N}=2\;A$	$R_{2}=0.27\;$Ω $I_{2N}=3\;A$	$R_{3}=0.16\;$Ω $I_{3N}=5\;A$
INMETRO [15]	$R_{1}=0.4\;$Ω $I_{1N}=2\;A$	$R_{2}=0.16\;$Ω $I_{2N}=5\;A$	$R_{3}=0.08\;$Ω $I_{3N}=10\;A$
FER [18]	$R_{1}=7.14\;$Ω $I_{1N}=0.1\;A$	$R_{2}=2.14\;$Ω $I_{2N}=0.3\;A$	$R_{3}=0.714\;$Ω $I_{3N}=1\;A$

**Table 2 sensors-23-08510-t002:** Sampling frequencies.

Frequency	Sampling Rate	Number of Samples
50 Hz	100 kSa/s	20,000
100 Hz	100 kSa/s	10,000
1 kHz	200 kSa/s	2000
3 kHz	201 kSa/s	670
5 kHz	200 kSa/s	400
10 kHz	200 kSa/s	200

**Table 3 sensors-23-08510-t003:** Phase error ϕ between the two NI PXI-4461 channels.

Frequency	Phase Error
50 Hz	−1 µrad
100 Hz	0 µrad
1 kHz	−8 µrad
3 kHz	6 µrad
5 kHz	−20 µrad
10 kHz	−31 µrad

**Table 4 sensors-23-08510-t004:** Phase displacement values—reference shunt.

Frequency	Phase Displacement	Measurement Uncertainty
50 Hz	0.3 µrad	0.7 µrad
100 Hz	0.8 µrad	0.7 µrad
1 kHz	7.1 µrad	0.7 µrad
3 kHz	21 µrad	2 µrad
5 kHz	36 µrad	4 µrad
10 kHz	71 µrad	7 µrad

**Table 5 sensors-23-08510-t005:** Phase displacement values—1 A FER shunt.

Relative Comparison Method					
Frequency					
50 Hz	100 Hz	1 kHz	3 kHz	5 kHz	10 kHz
2 µrad	1 µrad	−10 µrad	−7 µrad	−35 µrad	−70 µrad
**Absolute Phase Verification with Reference Shunt**					
Frequency					
50 Hz	100 Hz	1 kHz	3 kHz	5 kHz	10 kHz
0 µrad	0 µrad	7 µrad	32 µrad	41 µrad	72 µrad

**Table 6 sensors-23-08510-t006:** Absolute phase error between methods.

1 A Shunt					
Frequency					
50 Hz	100 Hz	1 kHz	3 kHz	5 kHz	10 kHz
2 µrad	1 µrad	17 µrad	39 µrad	76 µrad	143 µrad

**Table 7 sensors-23-08510-t007:** NI PXI-4461 phase error measurement uncertainty $u\left(\phi \right)$.

Frequency	Measurement Uncertainty
50 Hz	2 µrad
100 Hz	2 µrad
1 kHz	5 µrad
3 kHz	11 µrad
5 kHz	10 µrad
10 kHz	16 µrad

**Table 8 sensors-23-08510-t008:** Total measurement uncertainty—relative method.

Frequency	Measurement Uncertainty
50 Hz	8 µrad
100 Hz	8 µrad
1 kHz	18 µrad
3 kHz	28 µrad
5 kHz	34 µrad
10 kHz	57 µrad

**Table 9 sensors-23-08510-t009:** Total measurement uncertainty—absolute method.

Frequency	Measurement Uncertainty
50 Hz	5 µrad
100 Hz	5 µrad
1 kHz	12 µrad
3 kHz	23 µrad
5 kHz	21 µrad
10 kHz	34 µrad

## Data Availability

All data underlying the results are available as part of the article and no additional source data are required.

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
