# Peer review of "Evaluation of the Relative Phase Comparison Method at Its Limits Used for Absolute Phase Determination of an AC Cage Current Shunt Group"

_sensors, 2023, doi:10.3390/s23208510_

Round 1

Reviewer 1 Report

The previously described method from ref. [7] is applied to the set of shunts of interest of authors. The results are compared over the absolute phase measurement method. In this perspective the novelty of the paper is limited, though the results are reported correctly and may be accepted as the original contribution.

It could be elaborated more regarding the reason to select the method described in [7] for detail investigation.

Many references in the list are rather old. Despite the traditional field of electrical measurements references could be revised and more up to date research publications cited.

It could be commented more about how authors decide that the difference between results at higher frequencies is due to the errors of used instrumentation. Worse performance of the relative phase comparison method then is not proved to degrade at higher frequencies. It is just used instrumentation for the method testing exhibit higher errors. If testing setup errors are equal in the whole frequency range and difference is obvious at higher frequencies authors could conclude that the method from [7] is not good at higher frequencies. Therefore, I suggest to rethink the conclusions drawn.

I think the term “Referent shunt” should be substituted by “reference shunt”.

Author Response

The authors thank the reviewer for the review of the paper. We have carefully considered each of the comments and recommendations and have incorporated them into the revised manuscript. Thank you for the time and effort put into writing the review.

Reviewer 2 Report

The aim of the research presented in the article was to verify the method of determining the shunt phase error (absolute phase) by measuring the phase difference (relative phase) in a set of three shunts for nominal resistance ratios higher than those presented in the publication [7]. The authors tried to determine the limit of applicability of this method precisely due to the value of the ratios of the nominal resistance values of the three shunts forming the mentioned set. To verify the analyzed method, a direct comparison method with a reference shunt with known phase frequency characteristics was used.

The authors conducted interesting experiments, analyzed the results and concluded that the method based on the comparison of three shunts for the adopted set is not reliable above a frequency of 1 kHz. indeed, the difference in results obtained using the two methods presented increases significantly above 1 kHz. Perhaps the article should address the reasons for this discrepancy. Whether all uncertainty components were correctly identified in the experiments performed.

Referring to the above comment, it is not clear how the type B uncertainty given in Table 7 was determined. The sentence given in the article (below) does not explain how this uncertainty was determined.

The measurement uncertainties depend 300 on the repeatability of the measurements and the uncertainty associated with the referent 301 current shunt taken from its calibration certificate.   

Another example. Comparing 7.14W and 0.714W shunts at a current of 0.1 A, we compare voltages of 714 mV and 71.4 mV. How were the measurement ranges of the NI4461 card selected? Did the fact that when using a differential input the value of the CMRR coefficient significantly decreases with increasing frequency affect the accuracy of the phase difference measurement?

There are incorrect resistor symbols in Table 1 - they are all marked as R1 and errors in the designations of nominal currents.

Author Response

(The authors gave the same response as above.)
